# Aphid Resistance in *Pisum* Affects the Feeding Behavior of Pea-Adapted and Non-Pea-Adapted Biotypes of *Acyrthosiphon pisum* Differently

**DOI:** 10.3390/insects13030268

**Published:** 2022-03-08

**Authors:** Mauricio González González, Jean Christophe Simon, Akiko Sugio, Arnaud Ameline, Anas Cherqui

**Affiliations:** 1UMR CNRS 7058 EDYSAN (Écologie et Dynamique des Systèmes Anthropisés), Université de Picardie Jules Verne, 33 Rue St Leu, 80000 Amiens, France; mau.g.g@outlook.com (M.G.G.); arnaud.ameline@u-picardie.fr (A.A.); 2IGEPP, INRAE, Institute Agro, University Rennes, 35653 Le Rheu, France; jean-christophe.simon@inrae.fr (J.C.S.); akiko.sugio@inrae.fr (A.S.)

**Keywords:** *Acyrthosiphon pisum*, peas, resistant/susceptible genotype, EPG, body mass

## Abstract

**Simple Summary:**

Resistance of a *Pisum fulvum* and five *Pisum sativum* genotypes to *Acyrthosiphon pisum* pea and alfalfa-adapted biotypes was evaluated by measuring aphid body mass, confirming the variable resistance level of these genotypes. The feeding behavior of the aphids on the *Pisum* genotypes was then examined by electropenetrography (EPG). The EPG results suggested that the resistance of *Pisum* genotypes to non-adapted *A. pisum* resides in mesophyll and phloem tissues while the resistance variation of *P. sativum* to pea adapted aphids may be influenced by the quality of phloem sap.

**Abstract:**

Resistant genotypes of crops have emerged as an alternative and sustainable solution to pesticide use against pest insects. The resistance depends on the genetic diversity of the host plant and the pest species and can cause an alteration of the insect behavior. The aim of this work was to characterize the resistance level of different *Pisum* genotypes (one *P. fulvum* and five *P. sativum* genotypes) to two biotypes of the aphid *Acyrthosiphon pisum,* respectively adapted to pea and alfalfa, by measuring the individual aphid weight and analyzing aphid feeding behavior by electropenetrography (EPG). Aphid body mass was influenced by *Pisum* genotypes reflecting variation in their resistance level. *P. fulvum* was the most resistant to the *A. pisum* pea biotype (ArPo28 clone) and showed intermediate resistance to the *A. pisum* alfalfa biotype (LSR1 clone). The resistance levels of the five *P. sativum* genotypes to the two aphid biotypes were variable and more pronounced for the alfalfa biotype. EPG data showed that ArPo28 on *P. fulvum* and LSR1 on all the *Pisum* genotypes spent shorter time phloem feeding compared to ArPo28 on *P. sativum* genotypes, indicating that the resistance of *Pisum* genotypes to non-adapted *A. pisum* resides in mesophyll and phloem cells. In the meantime, ArPo28 on *P. sativum* genotypes with a different level of resistance spent a similar length of time phloem feeding, indicating that the quality of phloem sap of the resistance genotypes may not be optimal for the aphid. The study indicated that the resistance of *Pisum* genotypes to the two *A. pisum* biotypes involves different genetic factors and mechanisms that affect the aphid differently.

## 1. Introduction

Each year, aphids inflict serious economic problems in agriculture. Aphids are sap-feeding pests of many crops and are mainly controlled by a range of insecticides; however, due to the toxicity of the pesticides to beneficial insects, the recurrent development of insecticide resistance and the current EU moratorium on the use of certain insecticides, more environmentally sound pest control strategies are required. Although the deployment of insect-resistant crops has great potential to reduce pesticide use, the mechanisms of plant resistance to aphids are not well understood. The only cloned and characterized genes known to confer resistance against aphids encode NBS-LRR Vat in melon [1] and Mi in tomato [2]. The corresponding aphid avirulence proteins, which are presumed to be salivary proteins, have yet to be identified, resulting in a considerable gap in our knowledge of how plants sense an aphid attack and trigger defense reactions. One approach to characterize plant barriers responsible for aphid resistance lies in the electrical penetration graph (EPG), which tracks aphid feeding behavior during interaction with the plant [3,4]. Behavior of *Aphis gossypii* on the melon with and without *Vat* was studied using EPG and revealed a significant shortening of the phloem-feeding phase on the resistant plants [5,6]. In addition, the aphid took longer time to reach phloem and to salivate [5]. The results indicate that the resistance factors expressed by *Vat* reside in the mesophyll and phloem cells of melon. Similarly, *Macrosiphum euphorbiae* spends longer time to reach phloem and less time to feed on phloem sap in the resistant tomato carrying the *Mi-1.2* gene, compared to when it is on the susceptible one [7]. The authors also report behavioral variation between two different *M. euphorbiae* clones.

Non-host resistance against parasites may involve multiple genes and the mechanism of resistance varies depending on the interaction [8]. Aphid behaviors on Arabidopsis and barley were studied by Escudero-Martinez et al. [9] using two aphid species, *Myzus persicae* (host: *Arabidopsis*; poor-host: barley) and *Rhopalosiphum padi* (host: barley; non-host: *Arabidopsis*). In this study, *R. padi* failed to reach the phloem of non-host *Arabidopsis*, while *M. persicae* reached phloem but failed to sustain phloem-feeding on poor-host barley, indicating two different resistance mechanisms residing in different plant tissues.

In this study, we focused on the interactions between the *Pisum* species and the pea aphid, *Acyrthosiphon pisum*. *A. pisum* forms at least 15 biotypes, each of which is specialized to feed on one or a few legume species [10,11]. The *A. pisum* pea-adapted biotype is a major pest of the cultivated pea, *P. sativum. P. sativum* is an autogamous diploid plant and a large amount of effort has been made to improve its resilience to biotic and abiotic stressors. In this context, a core collection of 240 genotypes of pea and wild relative accessions representative of worldwide *Pisum* diversity has been created [12]. In a previous study, we conducted an analysis on the interaction between the 240 genotypes and *A. pisum* clones. Our fecundity assay using an *A. pisum* clone adapted to pea (pea biotype, ArPo28 clone) and alfalfa (alfalfa biotype, LSR1 clone) revealed a continuum in plant susceptibility and resistance [13]. None of the pea genotypes were totally resistant to the *A. pisum* pea biotype, while none of the pea genotypes were totally susceptible to the *A. pisum* alfalfa biotype. Nonetheless, the pea fecundity scores of the two biotypes were correlated, suggesting that the same resistance mechanisms operate against the two biotypes of *A. pisum*. Interestingly, three *P. fulvum* genotypes showed the highest level of resistance to the *A. pisum* pea biotype and partial susceptibility to the alfalfa biotype.

Previously, *A. pisum* behavior on adapted and non-adapted legume plants was examined using an EPG technique, although the recording period was relatively short (4 h) [14]. In this work, the authors showed that *A. pisum* clones adapted to alfalfa cannot establish phloem feeding on pea, while pea-adapted clones spend longer time feeding on phloem sap. The results indicate that the pea resistance mechanism against non-adapted biotypes resides in phloem cells. We hypothesized that the variable resistance levels we observed among the core collection of pea genotypes might be the consequence of the differences in feeding behavior of *A. pisum*. To evaluate the effect of the *Pisum* resistance on the behavior of the two *A. pisum* biotypes and to locate where the resistance factors reside, we selected from the 240 pea genotypes five *P. sativum* genotypes and a *P. fulvum* genotype showing contrasted resistance levels to the two *A. pisum* biotypes, confirmed the variation in resistance levels by using another measurement of aphid performance, and examined feeding behavior of pea- and alfalfa-adapted aphids with EPG. 

## 2. Materials and Methods

### 2.1. Plants and Insects

Five genotypes of *Pisum sativum* (AeD99OSW- 49-5-7, NEVE, CAMEOR, CE101 = FP and CHEROKEE) and one genotype of *Pisum fulvum* (WT304) were studied. These genotypes showed contrasted differences in resistance levels against one or the other *A. pisum* biotypes in our previous work [13]. Details about these six *Pisum* genotypes are given in Appendix A. Plants used for the experiments were grown in a phytotron with the following temperature, relative humidity, and light conditions: 20 ± 1 °C, 60 ± 5%, and 16:8 (hours of light:dark) at 4.7 klux. 

The two biotypes of *A. pisum*, a pea-adapted biotype, the ArPo28 clone, and an alfalfa-adapted biotype, the LSR1 clone, were provided by INRAE Le Rheu [13]. These aphid clones are deprived of any secondary symbionts reported in *A. pisum*. For each aphid biotype, colonies were generated from a single apterous parthenogenetic female and continuously reared on broad bean plants (*Vicia faba* “Castell”) in a ventilated Plexiglas^®^ cage in different growth chambers under 20 ± 1 °C, 60 ± 5% R.H., and 16:8 (L:D) photoperiod. A synchronization process was established to produce *A. pisum* individuals of the same age (8 ± 1 days old) by isolating 15 adult aphids on 3-week-old broad bean plants. Each plant was isolated with an individual breathable plastic cover. After 24 h, the adults were removed and only neonate nymphs were left during the 8 days. 

### 2.2. Aphid Body Mass Measurements on the Six Pisum Genotypes

In order to confirm the resistance levels assessed in the previous study by a fecundity assay, we measured as a proxy the aphid body mass produced on each of the six *Pisum* genotypes used in this work. For that, 10 adult aphids from each aphid clone were installed on the different genotypes of *P. sativum* and *P. fulvum*. After 24 h, adults were removed and only their offspring were kept on the plant. Each plant was then isolated with a breathable plastic cover and grown in a climatic chamber under the conditions described above. Four replicates were performed for each plant genotype. After 8 days, all surviving aphids (ranging from 12 to 80 depending on the aphid-plant interaction) were collected from each plant and stored in Eppendorf tubes in a freezer at −80 °C. Each aphid of the 12 conditions (2 aphid biotypes × 6 plant genotypes) was then individually weighed using an electronic precision balance (Mettler M3, class 1, Max: 3 g Low: 1 µg, T = −3G [dd] = 1 µg).

### 2.3. Aphid Feeding Behavior Monitored by EPG

The feeding behavior of the two biotypes of *A. pisum* on the six *Pisum* genotypes was studied using the electropenetrography technique, EPG [15,16]. Eight hours of recordings were carried out in the middle of the 16 h of photophase under controlled conditions (24 ± 1 °C, 60 ± 5% RH, and 16L:8D photoperiod, 2.5 klux). EPG recordings were obtained according to a set-up consisting in sticking a thin gold wire (Ø 20 μm and 2 cm long) with a conductive silver glue (EPG systems, Wageningen, the Netherlands) on the dorsal part of the aphid’s abdomen, which was connected to an electrical closed circuit comprising the aphid and its host plant. Aphids were placed on the abaxial part of a fully expanded leaf of 3-week-old plants. An eight-channel set-up allowed a simultaneous acquisition of the feeding behavior of eight aphids. Acquisition and analysis of the EPG waveforms were carried out with EPG Stylet + software (EPG Systems, www.epgsystems.eu, accessed on 10 January 2022). EPG waveforms can be related to three major aphid activities: non-probing where aphid stylets are outside plant tissues (NP), probing in epidermis or mesophyll (P), and probing in vascular tissues (E). Probing in non-vascular tissues included typical pathway activity that represents the progressive movement of stylets within the apoplast accompanied by short punctures into cells adjacent to the stylet track (C waveform) and derailed stylet activities (F waveform). Probing in vascular tissues embraced the ingestion of xylem sap (G waveform), salivation into sieve elements (E1 waveform), and ingestion of phloem sap from sieve elements (E2 waveform).

Eight EPG parameters were calculated with EPG-Calc 6.1.7 software [17] to assess the aphid feeding behavior: the total duration of non-probing (s_NP), the total duration of probing (s_Pr), and the time to first stylet penetration within plant tissues (t > 1 Pr). We also evaluated the total duration of pathway phase (s_C), the total duration of phloem salivation (s_E1), and the total duration of phloem sap ingestion (s_E2). Finally, the total duration of derailed stylet phase (s_F) was also calculated. The feeding behavior of 32 aphids was recorded for each of the 12 conditions (2 aphid clones × 6 plant genotypes). Each plant and aphid were used only once.

### 2.4. Statistical Analysis

All statistical analyses were performed using the Rstudio software version 1.4.1717 (The R Foundation, https://www.r-project.org/, accessed on 30 January 2022). Data on aphid body mass were analyzed using a one-way ANOVA, and post-hoc comparisons using DGC test (Di Rienzo, Guzman, Casanoves), *p* < 0.05. EPG-derived parameters were analyzed using a non-parametric analysis of variance (Kruskal-Wallis), since their distribution did not follow normal law. Pairwise comparisons were made when the *p* value was inferior to 0.05, 0.01, or 0.001. For some parameters, the Mann–Whitney U rank sum test was carried out to point out differences between *Pisum* genotypes for each aphid biotype. To visualize how the different *Pisum* genotypes influence the feeding behavior of the pea- and alfalfa-adapted *A. pisum* clones, we conducted a principal component analysis (PCA) on a selection of EPG parameters that exhibit significant differences between *Pisum* genotypes. PCA was run for each biotype separately using the FactoMineR package. Axes with an eigenvalue >1 were considered and the first two dimensions with the highest loadings being selected.

## 3. Results

### 3.1. Variation in Aphid Body Mass on Pisum Genotypes 

Significant differences were found in the larval weight of the *A. pisum* pea biotype (ArPo28 clone) reared on the different *Pisum* genotypes (Figure 1). The nymph reared on the *P. fulvum* genotype, WT304, exhibited the lowest body mass (0.91 ± 0.37 mg). Nymphs on *P. sativum* genotypes AeD99OSW- 49-5-7 and NEVE had lower body mass (1.09 ± 0.39 mg and 1.21 ± 0.40 mg respectively) than CE101 = FP, CAMEOR and CHEROKEE (1.81 ± 0.27 mg, 1.62 ± 0.34 mg and 1.51 ± 0.41 mg, respectively). 

Significant differences were found in the larval weight of *A. pisum* alfalfa biotype (LSR1 clone) reared on the different *Pisum* genotypes (Figure 2). No surviving nymph was observed on two *P. sativum* genotypes (NEVE and AeD99OSW- 49-5-7), and the lowest larval body mass was observed on *P. sativum* genotype CAMEOR (0.32 ± 0.09 mg). The nymph with the highest body mass was observed on CE101 = FP genotype (0.59 ± 0.18 mg) while CHEROKEE (0.49 ± 0.18 mg) and the *P. fulvum* genotype WT304 (0.49 ± 0.18 mg) displayed similar intermediate body mass.

### 3.2. Difference in Feeding Behavior between A. pisum Biotypes on the Six Pisum Genotypes

The feeding behavior of the *A. pisum* pea biotype, ArPo28, was influenced significantly by the different *P. sativum* and *P. fulvum* genotypes for the following parameters (Table 1): the total duration of non-probing (*p* = 0.018), the total duration of probing (*p* = 0.016), the total duration of the pathway phase (*p* = 0.003), the total duration of the phloem ingestion phase (*p* < 0.001), and the time to the first phloem phase (*p* = 0.046). 

Most of the variation was due to the differences of aphid feeding behavior between *P. sativum* and *P. fulvum*. The total duration of non-probing (s_NP) was significantly longer on *P. fulvum* WT304 (86 ± 9 min) compared with AeD99OSW- 49-5-7 (47 ± 7 min), CAMEOR (53 ± 11 min), and CE101 = FP (55 ± 12 min). Inversely, the total duration of probing (s_Pr) was significantly shorter on WT304 compared to all the *P. sativum* genotypes except NEVE. Total duration of the pathway phase (s_C) and, consequently, the time spent to access to phloem vessels were significantly longer on *P. fulvum* compared with each of the five *P. sativum* genotypes. Concerning the phloem phase parameters, the total duration of the phloem ingestion phase (s_E2) was significantly shorter (64 ± 10 min) on *P. fulvum* in comparison to each of the five *P. sativum* genotypes. There was no significant difference for the total duration of stylet derailment (s_F) between genotypes of *P. sativum* and *P. fulvum*. The results indicate a clear behavioral difference of ArPo28 between the two *Pisum* species. The feeding behavior of ArPo28 on resistant and susceptible pea cultivars was not significantly different, except that resistant genotype NEVE, but not AeD99OSW- 49-5-7, showed shorter probing time compared to other *P. sativum* genotypes.

The different *Pisum* genotypes had a significant effect on the feeding behavior of the *A. pisum* alfalfa biotype (LSR1 clone) for the following parameters (Table 2): the total duration of non-probing (*p* < 0.001), the total duration of probing (*p* < 0.001), the total duration of pathway phase (*p* = 0.008), and the total duration of derailed stylet phase (*p* = 0.010).

The total duration of probing (s_Pr) was significantly shorter on two resistant genotypes, NEVE and AeD99OSW- 49-5-7, and inversely the total duration of non-probing was significantly longer for these two genotypes compared to the rest of genotypes except CHEROKEE. LSR1 on CHEROKEE, CE101 = FP, and CAMEOR showed the longest probing time and LSR1 on CE101 = FP and CAMEOR showed the shortest non-probing time. LSR1 aphids on *P. fulvum* WT304 showed intermediate probing and non-probing times. The total duration of the pathway phase (s_C) was significantly longer on *P. fulvum* WT304 (196 ± 24 min), NEVE (171 ± 17 min), AeD99OSW- 49-5-7 (152 ± 19 min), and CHEROKEE (144 ± 16 min) than on CE101 = FP (101 ± 18 min) and was intermediate on CAMEOR. For the three parameters related to the phloem phase (E), no statistical difference was observed as the numbers of aphids that reached phloem were low in all the genotypes. The percentage of aphids exhibiting phloem ingestion activity varied from 0% for NEVE and 8% for AeD99OSW- 49-5-7 to 17–30% for CAMEOR, CHEROKEE, CE101 = FP and *P. fulvum* WT304. Although only 22% of aphids (four out of 18 aphids studied) reached phloem on WT304, they spent longer time feeding on phloem (56 ± 34 min) compared to other genotypes. Finally, the total duration of stylet derailment (s_F) was significantly longer on CAMEOR (284 ± 26 min) compared with NEVE (185 ± 23 min) and WT304 (185 ± 32 min). 

The principal component analysis (PCA) based on the significant EPG parameters related to pathway phase and phloem activities showed clear differences in the feeding behavior of the pea-adapted biotype between the *P. fulvum* and *P. sativum* genotypes (Figure 3A). The weights (loadings) of the six retained EPG parameters (total duration of non-probing, probing, pathway, salivation, ingestion, and time to first phloem phase) on the two first axes are shown in Figure 3B. The first two components (PC1 and PC2) resumed 69.4% of the total inertia (i.e., total variance of dataset).

PCA on the alfalfa biotype EPG dataset showed a clear distinction between the *P. sativum* genotypes on the first axis, with resistant genotypes NEVE and AeD99OSW- 49-5-7 separated from more susceptible ones, CE101 = FP and CAMEOR (Figure 4B). The second axis tended to separate aphid feeding behaviors on *P. fulvum* from *P. sativum*, although there was some overlap between these two. The weights (loadings) of the five retained EPG parameters (total duration of non-probing, probing, pathway, derailment, and ingestion) on the two first axes are shown in Figure 4B. The first two components (PC1 and PC2) resumed 82.8% of the total inertia. 

### 3.3. Comparison of the Feeding Behavior between the Two Aphid Biotypes

Comparison of the feeding behavior of the two aphid clones, ArPo28 and LSR1, showed no difference on *P. fulvum* (Figure 5), which is not a suitable host plant for either of the aphid clones as indicated by the short proportion of feeding activity in the E2 phase (phloem ingestion). On all the *P. sativum* genotypes, a significantly longer E2 phase (phloem ingestion, *p* < 0.001) and significantly shorter F phase (derailed stylet, *p* < 0.001) of ArPo28 compared to LSR1 were observed. Also, the non-probing time of ArPo28 was significantly shorter (*p* < 0.01) compared to LSR1 on all the *P. sativum* genotypes except CAMEOR and CE101 = FP.

## 4. Discussion

In a previous study, we screened a collection of 240 *Pisum* genotypes for aphid resistance and found variation in resistance/susceptibility levels in both interactions with a pea-adapted and an alfalfa-adapted biotype of the *A. pisum* complex. The aim of the present study was to examine the resistance levels of *Pisum* genotypes and their influence on aphid feeding behavior and to identify in which plant-cell layers the resistance factors may reside.

As a first step of this study, we measured the body mass of individual aphids feeding on the different *Pisum* genotypes to confirm the resistance ranking obtained with a fecundity assay in an earlier study [13] (Appendix A). Body mass and fecundity scores are commonly used to screen for plant resistance as they best capture the impact of different plant resistance on aphid fitness [18]. Overall, we found good agreement in the ranking of resistance/susceptibility levels of the six *Pisum* genotypes between the two types of measurements. Body mass measurements identified a group of *Pisum* genotypes (WT304, AeD99OSW-49-5-7, NEVE) partially resistant to the pea-adapted clone, ArPo28, with *P. fulvum* WT304 being the most resistant as in the ranking based on fecundity score. This group differs from another formed by susceptible genotypes (CAMEOR, CE101 = FP, and CHEROKEE), which were also characterized as susceptible in the fecundity assay (except CAMEOR, which shows an intermediate phenotype). Resistance ranking to the alfalfa-adapted clone, LSR1, was also consistent between the measurements. Two genotypes (AeD99OSW-49-5-7 and NEVE) were completely resistant because aphids could not develop on them, so the aphid body mass was zero. These genotypes correspond to the ones with the lowest fecundity scores. The third most resistant genotype (CAMEOR) was the same in both types of measurements. The ranking for the last three was slightly different, which may reflect some trade-off between the two traits [19,20]. When the body mass of the two aphid clones is compared, ArPo28 was always heavier than LSR1, with a three- to six-fold difference for *P. sativum* genotypes and two-fold for *P. fulvum*. This confirms the overall better adaptation of the pea biotype ArPo28 on *P. sativum* [21]. It should be recalled here that the fecundity scores from the previous study [13] were obtained using a different protocol for ArPo28 and LSR1 (fecundity was measured from three first instar aphids for ArPo28 while it was measured from 10 mixed-age aphids for LSR1) so that direct comparison was not possible. 

The EPG analysis showed no difference in the feeding behavior of ArPo28 between the *P. sativum* genotypes although they showed contrasting levels of resistance. Within 90% of the probing time, ArPo28 spent 50% of its time in ingestion of sieve elements sap, which is normally found when *A. pisum* biotypes feed on their respective preferred legume hosts [14]. This indicates that the factors associated with the partial resistance of NEVE and AeD99OSW-49-5-7 probably reside in the phloem and likely result from the ingestion of a constitutive or induced toxic compound by the aphid. It has been shown in legumes that some compounds such as saponins, phenolic compounds, and flavonoid glycosides reduce aphid fecundity [22,23,24,25]. Whether NEVE and AeD99OSW-49-5-7 produce higher levels of these toxic compounds requires further investigation, but based on our body mass and EPG data, we can speculate that the resistance mechanism of these genotypes is antibiosis.

We found that the feeding behavior of ArPo28 on *P. fulvum* was impaired, and this result, with the performance data (i.e., body mass and fecundity scores), suggest that this plant is not a suitable host for the pea-adapted ArPo28. The time to first penetration (t > 1 P) was not impacted, suggesting no epidermal resistance nor involvement of surface compounds such as volatile organic compounds (VOCs) or waxes [26]. However, ArPo28 on *P. fulvum* spent longer non-probing time and shorter probing time compared to some *P. sativum* genotypes. Access to the phloem was significantly delayed compared to feeding on *P. sativum*. Long pathway periods in the mesophyll are generally related to a chemical resistance associated with deterrent compounds present in the mesophyll [9], or physical resistance related to the structure or thickness of the mesophyll cell walls or those of the cribbed tubes of the phloem [27]. These long mesophyll periods would delay the access of the aphid stylets to the phloem and were reported in EPG studies on different aphid/plant systems [28,29,30]. Furthermore, phloem sap ingestion time of ArPo28 on *P. fulvum* was three times shorter compared to *P. sativum* genotypes, which could be explained either by mesophyll or phloem resistance factors or by factors related to the chemical composition of the sap. 

The feeding behavior of the alfalfa-adapted LSR1 on the various *P. sativum* genotypes was very different from the one of ArPo28, confirming a previous EPG work on three *A. pisum* biotypes on their respective host and non-host legumes [14]. Globally, LSR1 spent much longer time in the mesophyll and much less time in the phloem than ArPo28, whatever *Pisum* genotype was tested. As a result, LSR1 ingested little or no phloem sap, the duration was short, and above all, the number of aphids reaching this phase was limited (0% to 30%). The derailment activity was also much more pronounced in LSR1, with durations of probing time higher than 40%. This could be due to maladapted aphid stylets or to the physical structure of the plant tissue, suggesting that alfalfa aphids encounter resistance in the mesophyll. The increased duration of derailment of aphid stylets is often observed on resistant plants [16], on virus-infected plants [31], and in aposymbiotic aphids [32]. Throughout the navigation of the stylets in plant tissues to reach the sieve element, the aphids excrete a gelling saliva that envelops the stylets along the track in the mesophyll [33]. The incapability to form a hardened sheath and the diffusion of its components into the apoplastic medium could lead to the loss of the main functions of the sheath including lubrication, cell wall digestion and detoxification [34]. The derailment of the LSR1 in the mesophyll is likely due to the failure of mechanical penetration of the stylet that could result from the production of maladapted aphid sheath saliva. The composition of the sheath saliva may differ between pea- and alfalfa-adapted aphids and explain the maladaptation of the LSR1 stylets to these *Pisum* genotypes. Proteomic and transcriptomic analyses were conducted to study the composition of pea aphid saliva in order to determine salivary effectors related to the specificity of these aphids to their host plant [35,36,37]. Although expression differences in salivary genes were found between pea and alfalfa biotypes [35], whether these genes encode gelling or aqueous salivary proteins is unknown. Comparing the sheath saliva of pea-adapted and non-adapted biotypes may help to understand the basis of non-host resistance in the *Pisum* genotypes to alfalfa-adapted *A. pisum* clones.

Although the feeding behavior of LSR1 on *Pisum* genotypes did not show clear difference when EPG parameters were analyzed one by one, the multivariate analysis showed that aphids on the two most resistant genotypes (NEVE and AeD99OSW-49-5-7) were grouped together and distinguished from more susceptible ones. The PCA also showed some separation of the feeding behavior of LSR1 on *P. fulvum*, suggesting different resistance factors in this plant species. It is noteworthy that the behaviors of ArPo28 and LSR1 on *P. fulvum* were very similar, while these two clones showed contrasting behaviors on *P. sativum* (Figure 5). This indicates that although *P. fulvum* is a close relative of *P. sativum,* it is a similarly poor host for both biotypes.

Our results thus show that resistance mechanisms of different plant species from the same genus can reside in different plant tissues and affect the behavior of non-adapted aphids differently, a result in line with previous studies on non-host resistance factors compared between plants from different families [9].

## 5. Conclusions

The results of this work allow us to conclude that the resistance of the tested genotypes of *P. sativum* would not be explained by a mechanism of feeding disruption for the pea biotype of *A. pisum*. On the contrary, the generalized resistance that *P. sativum* displayed against the alfalfa biotype of *A. pisum* is thoroughly associated to a probing disruption located at the mesophyll and phloem level, which could result in the low larval development expressed as body mass measurements after 8 days. This indicates that the plant resistance factors of these genotypes affected each aphid biotype differently. 

## Figures and Tables

**Figure 1 insects-13-00268-f001:**
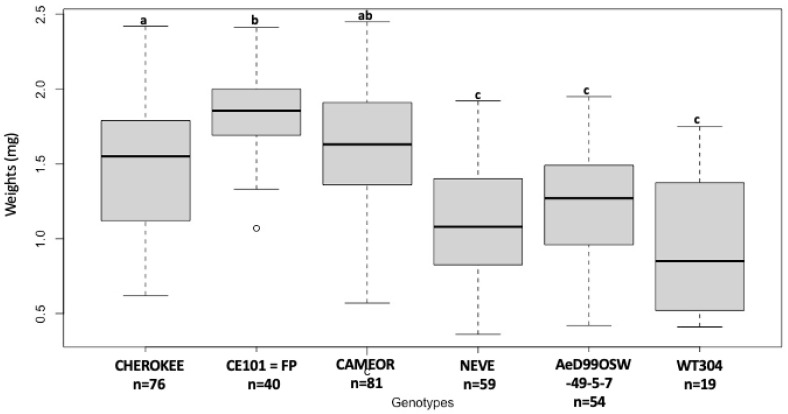
Larval body mass (8-day-old) of pea biotype, ArPo28, reared on different genotypes of *P. sativum* and on *P. fulvum.* Different letters indicate significant differences (one-way ANOVA, post-hoc comparisons using DGC, *p* < 0.05).

**Figure 2 insects-13-00268-f002:**
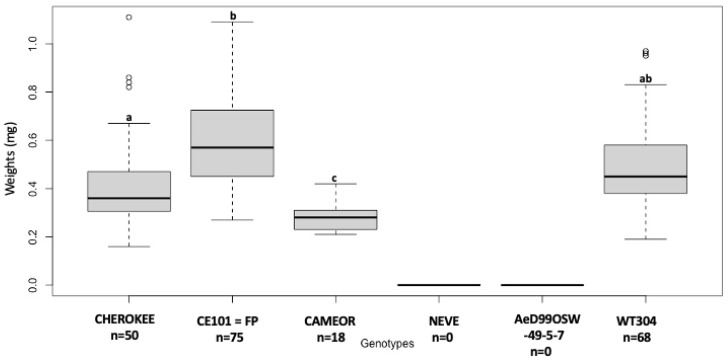
Larval body mass (8-day-old) of alfalfa biotype, LSR1, reared on different genotypes of *P. sativum* and on *P. fulvum.* Different letters indicate significant differences (one-way ANOVA, post-hoc comparisons using DGC, *p* < 0.05).

**Figure 3 insects-13-00268-f003:**
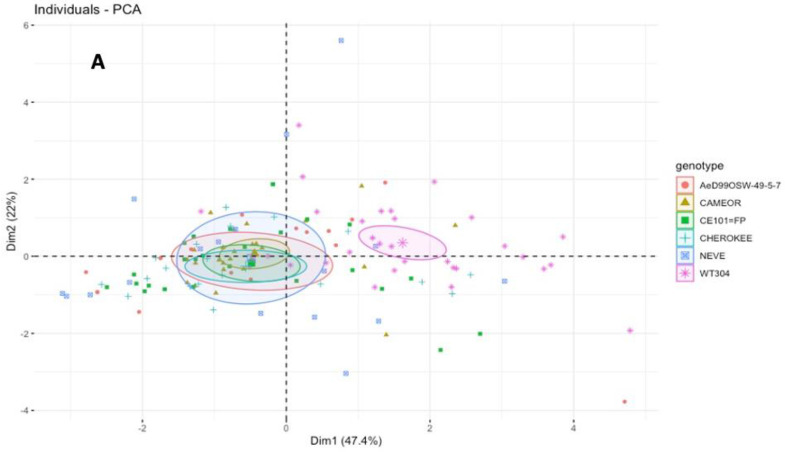
Principal component analysis (PCA) for the aphid trophic behavior by EPG study. Individuals factor map indicating the scores of aphids *A. pisum* feeding on six *Pisum* genotypes. The circles show the confidence ellipses (with a confidence level of 0.95) calculated with the barycenter of individuals for each parameter on the factorial plan. (**A**) PCA of individuals of pea biotype (ArPo28) probing on *Pisum* genotypes, (**B**) Variables factor map indicates the spatial ordination using five parameters: s_NP, total non-probing duration; s_C, total pathway phase; s_Pr, total probing duration; s_E1 total salivation duration; s_E2, total phloem ingestion duration; and t > 1 E, time to first phloem phase.

**Figure 4 insects-13-00268-f004:**
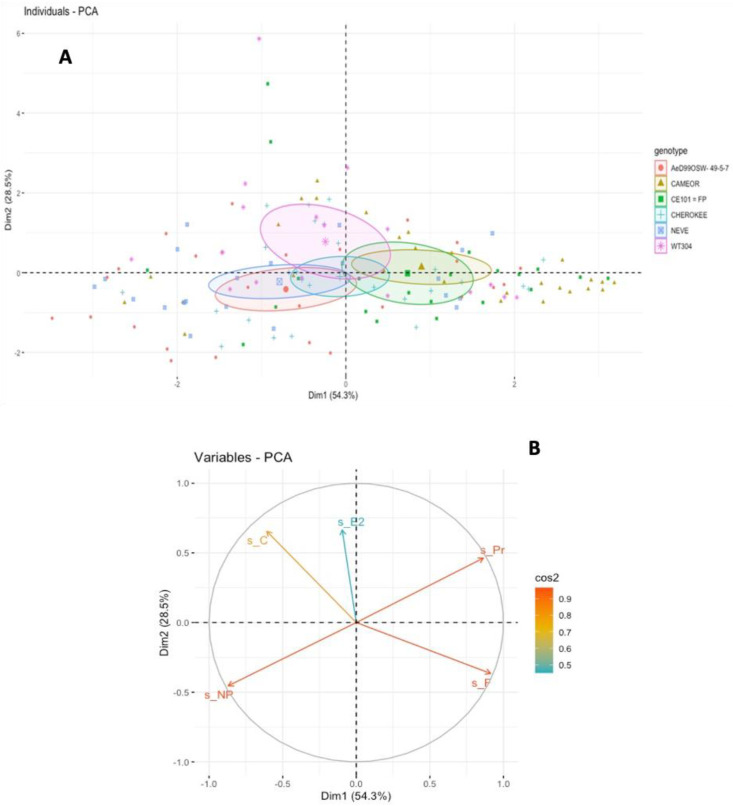
Principal component analysis (PCA) for the aphid trophic behavior by EPG study. Individuals factor map indicating the scores of aphids *A. pisum* feeding on six *Pisum* genotypes. The circles show the confidence ellipses (with a confidence level of 0.95) calculated with the barycenter of individuals for each parameter on the factorial plan. (**A**) PCA of individuals of alfalfa biotype (LSR1) probing on *Pisum* genotypes. (**B**) Variables factor map indicates the spatial ordination using five parameters: s_NP, total non-probing duration; s_C, total pathway phase; s_Pr, total probing duration; s_F, total stylets derailment duration; and s_E2, total phloem ingestion duration.

**Figure 5 insects-13-00268-f005:**
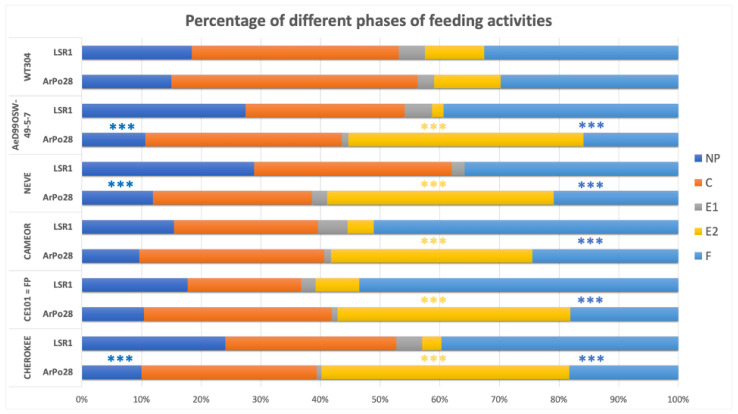
Percentage of the different phases of feeding activities during 8 h EPG recordings of pea biotype (ArPo28) and alfalfa biotype (LSR1) on *P. fulvum* (WT304) and on the different *P. sativum* genotypes (NP, non-probing; C, pathway; E1, phloem salivation; E2, phloem sap ingestion; F, derailment). Pairwise comparisons using Wilcoxon rank sum test with continuity correction were carried out (Mann–Whitney U test) and asterisks indicate a significant difference, **** p <* 0.001.

**Table 1 insects-13-00268-t001:** Electrical penetration graph parameters (means ± SEM) calculated for *A. pisum* pea biotype (ArPo 28 clone) during an 8 h monitoring session on different genotypes of *P. sativum* and on *P. fulvum*.

Pea Aphid ArPo28		*Pisum fulvum*	*Pisum sativum*				
	Kruskal-Wallis Test H(*P*)	WT304	AeD99OSW- 49-5-7	NEVE	CAMEOR	CE101 = FP	CHEROKEE
** *General phases* **							

**Total duration of non-probing**							
**(min. s_NP)**	13.56 (*)	86 ± 9 **a**	47 ± 7 **b**	58 ± 13 **ab**	53 ± 11 **b**	55 ± 12 **b**	54 ± 11 **ab**
***n* =**		*31*	*20*	*22*	*28*	*29*	*22*
**Total duration of probing**							
**(min. s_Pr)**	13.97 (*)	394 ± 10 **a**	422 ± 15 **b**	414 ± 14 **ab**	427 ± 11 **b**	425 ± 12 **b**	426 ± 10 **b**
***n* = **		*31*	*20*	*22*	*28*	*29*	*22*
**Time to first probe**							
**(min. t > 1 Pr)**	1.52 (NS)	12 ± 3	9 ± 3	18 ± 9	11 ± 5	14 ± 4	15 ± 5
***n* = **		*31*	*20*	*22*	*28*	*29*	*22*
** *Pathway phases (C)* **							

**Total duration of pathway phase**							
**(min. s_C)**	17.64 (**)	237 ± 15 **a**	180 ± 21 **b**	148 ± 24 **b**	171 ± 16 **b**	167 ± 12 **b**	159 ± 16 **b**
***n* = **		*31*	*20*	*22*	*28*	*29*	*22*
** *Phloem phases (E)* **							

**Total duration of salivation phase (min. s_E1)**	7.59 (NS)	16 ± 3	6 ± 1	14 ± 6	6 ± 1	5 ± 0	4 ± 1
***n* = **		*25*	*17*	*18*	*25*	*26*	*20*
**Total duration of phloem ingestion (min. s_E2)**	35.04 (***)	64 ± 10 **a**	215 ± 28 **b**	211 ± 31 **b**	186 ± 17 **b**	207 ± 21 **b**	225 ± 25 **b**
***n* = **		*22*	*17*	*18*	*24*	*26*	*19*
**Time to first phloem phase**							
**(min. t > 1 E)**	11.30 (*)	281 ± 30 **a**	168 ± 31 **b**	164 ± 28 **b**	226 ± 21 **b**	222 ± 21 **b**	192 ± 21 **b**
***n* = **		*23*	*17*	*17*	*24*	*26*	*19*
** *Other phases (F)* **							

**Total duration of derailed stylet phase (min. s_F)**	7.43 (NS)	171 ± 28	87 ± 22	116 ± 29	135 ± 24	96 ± 14	99 ± 23
***n* = **		*16*	*8*	*9*	*17*	*22*	*9*

Asterisks indicate a significant difference: * *p* < 0.05, ** *p* < 0.01, *** *p* < 0.001 associated with H (Kruskal–Wallis test); the letters indicate significant differences associated with following pairwise comparisons and *n* corresponds to the number of replicates. The Mann–Whitney U test was carried out between *Pisum* genotypes for the parameters s_NP and s_Pr.

**Table 2 insects-13-00268-t002:** Electrical penetration graph parameters (means ± SEM) calculated for *A. pisum* alfalfa biotype (LSR1 clone) during an 8 h monitoring session on different genotypes of *P. sativum* and on *P. fulvum*.

Alfalfa Biotype (LSR1)		*Pisum fulvum*	*Pisum sativum*				
	Kruskal-Wallis Test H(*P*)	WT304	AeD99OSW- 49-5-7	NEVE	CAMEOR	CE101 = FP	CHEROKEE
** *General phases* **							

**Total duration of non-probing (min.s_NP)**	22.43 (***)	103 ± 11 **ab**	151 ± 16 **a**	148 ± 16 **a**	77 ± 14 **b**	93 ± 13 **b**	120 ± 11 **ab**
***n* =**		*18*	*25*	*21*	*28*	*20*	*26*
**Total duration of probing**							
**(min. s_Pr)**	22.59 (***)	376 ± 11 **ab**	324 ± 17 **a**	331 ± 15 **a**	386 ± 13 **b**	402 ± 13 **b**	359 ± 11 **ab**
***n* = **		*18*	*25*	*21*	*28*	*20*	*26*
**Time to first probe**							
**(min. t > 1 Pr)**	3.09 (NS)	21 ± 7	23 ± 9	16 ± 6	18 ± 6	8 ± 2	17 ± 7
***n* = **		*18*	*25*	*21*	*28*	*20*	*26*
** *Pathway phases (C)* **							

**Total duration of pathway phase (min. s_C)**	15.66 (**)	196 ± 24 **a**	152 ± 19 **ab**	171 ± 17 **ab**	101 ± 18 **b**	122 ± 17 **b**	144 ± 16 **ab**
***n* = **		*18*	*25*	*21*	*28*	*20*	*26*
** *Phloem phases (E)* **							

**Total duration of salivation phase (min. s_E1)**	4.57 (NS)	25 ± 10	26 ± 12	11 ± 5	13 ± 4	25 ± 6	22 ± 5
***n* = **		*10*	*8*	*9*	*12*	*15*	*11*
**Total duration of phloem ingestion (min. s_E2)**	8.16 (NS)	56 ± 34	11 ± 3	-	39 ± 23	22 ± 5	16 ± 3
***n* = **		*4*	*2*	*0*	*5*	*6*	*6*
**Time to first phloem phase**							
**(min. t > 1 E)**	4.93 (NS)	421 ± 33	474 ± 5	-	283 ± 67	300 ± 63	347 ± 40
***n* = **		*4*	*2*	*0*	*5*	*6*	*6*
** *Other phases (F)* **							

**Total duration of derailed stylet phase (min. s_F)**	15.04 (*)	184 ± 32 **a**	224 ± 24 **ab**	185 ± 23 **a**	258 ± 29 **ab**	284 ± 26 **b**	200 ± 20 **ab**
***n* = **		*15*	*18*	*17*	*28*	*18*	*26*

Asterisks indicate a significant difference: * *p* < 0.05, ** *p* < 0.01, *** *p* < 0.001 associated with H (Kruskal–Wallis test); the letters indicate significant differences associated with following pairwise comparisons and n corresponds to the number of replicates.

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
