# Peer review of "Aphid Resistance in *Pisum* Affects the Feeding Behavior of Pea-Adapted and Non-Pea-Adapted Biotypes of *Acyrthosiphon pisum* Differently"

_insects, 2022, doi:10.3390/insects13030268_

Round 1
Reviewer 1 Report
The manuscript “Aphid resistance in Pisum affects the feeding behavior of pea adapted and non-pea-adapted biotypes of Acyrthosiphon pisum differently” by González and co-authors evaluated the resistance of five Pisum sativum and one Pisum fulvum genotypes to two biotypes of aphid Acyrthosiphon pisum by using the Electrical Penetration Graph (EPG) technique.
The work is well structured and presented. The introduction is clearly written and no inappropriate self-citations by Authors are present. It explain all the necessary information for understanding the other sections of this manuscript. The experimental plan is equally explained and methods are adequately reported. Results are properly reported and discussed.
However, I have few observations which should be considered by the authors in the submission of a revised version.
- The Figures and Tables are of low quality and hard to follow, and they should be improved.
- According to heading ”Comparison of the feeding behavior between the two aphid biotypes” in section 3.3 the feeding behavior (on the basis of five EPG parameters) of pea biotype (ArPo28) and alfalfa biotype (LSR1) was compared on each of 6 plant genotypes. Thus, the pairwise comparisons (t-Student or the Mann-Whitney U test ) should be made to clarify the differences (Figure 5).
- Line 232: insert ‘the’ after ‘showed’…. and CAMEOR showed longest probing time and….
- Line 340: Are long pathway periods related to “anti-repellent” or “repellent” compounds?
- Table S1 should be self-explanatory. Column 3 is “origin” not “origine”. It is not clear what have column 6 and 7 contained, since reference 13 has not been published yet. Please clarify.
As a whole I think that the manuscript is properly structured and written, methods and data are adequately reported and conclusions are supported by data so that I suggest to accept the manuscript after minor revision.
Author Response
1st Reviewer
- The Figures and Tables are of low quality and hard to follow, and they should be improved.
Authors’ response: We have made modifications for improving the quality of the figures and the clarity of the tables as mentioned by the reviewer.
- According to heading ”Comparison of the feeding behavior between the two aphid biotypes” in section 3.3 the feeding behavior (on the basis of five EPG parameters) of pea biotype (ArPo28) and alfalfa biotype (LSR1) was compared on each of 6 plant genotypes. Thus, the pairwise comparisons (t-Student or the Mann-Whitney U test ) should be made to clarify the differences (Figure 5).
Authors’ response: Mann-Whitney U test has been applied and some more significative differences were identified as shown in the new Figure 5.
- Line 232: insert ‘the’ after ‘showed’…. and CAMEOR showed longest probing time and….
Authors’ response: Done
- Line 340: Are long pathway periods related to “anti-repellent” or “repellent” compounds?
Authors’ response: correction was applied and “deterrent” word is used instead of “anti-repellent”.
- Table S1 should be self-explanatory. Column 3 is “origin” not “origine”. It is not clear what have column 6 and 7 contained, since reference 13 has not been published yet. Please clarify.
Authors’ response: the correction was made and the following text was added in the supplementary data
*ArPo28 aphid count. Three first instar (L1) larvae (generation 1, G1) produced on V. faba were installed on each Pisum genotype. Ten days after the G1 installation, three L1 larvae (G2) were installed on a new Pisum plant of the same genotype. Then, 18 days after the installation of G2 aphids, all their offspring (G3) were collected and counted [13].
**LSR1 aphid count. Ten mixed-aged aphids were installed on each Pisum plant and all aphids (G1 adults and G2 offspring) were collected and counted three weeks later [13].
Reviewer 2 Report
This manuscript reports the investigation results on resistant levels and mechanisms of 6 different Pisum genotypes to two biotypes of the aphid, Acyrthosiphon pisum, respectively adapted to pea and alfalfa, by measuring the individual aphid weight and by analysing aphid feeding behavior with EPG. Generally the experimental design, result presentation, as well as English writing, are well done, but a few concerns should be addressed before it can be accepted for publication:
1 About EPG experiments: How many replicates were used for recordings? What status of plants were used ( how high or with how many leaves)? Did the aphids and plants were used only once (used ones were discarded after each recordings) or used repeatedly? And, the word "parameter" now is replaced by "variables".
2 About results: image quality of Figures 1 to 4 and Tables 1 to 2 should be improved. Titles of Table 1 and Table 2 can be shortened by moving the notes in the titles to the end of the tables.
3 Discussion: the last paragraph of discussion part can be expanded a little to provide not only the general conclusion but also some detailed information of summary of results.
Author Response
2nd Reviewer
- About EPG experiments: How many replicates were used for recordings? What status of plants were used (how high or with how many leaves)? Did the aphids and plants were used only once (used ones were discarded after each recordings) or used repeatedly? And, the word "parameter" now is replaced by "variables".
Authors’ response: 32 replicates were done for the EPG recordings (line 166-168, of the revised version), but only n replicates (mentioned in tables) were then used for the EPG analysis and comparison of the parameters. The differences are due to some troubleshooting during the EPG recording (reduced total recording ie: less than 7h, escaped aphids, non-feeding activity recorded….)
The plants used are 3-week-old, and we used only 3rd and 4th leaf level from the apex.
Each plant and aphid were used only once.
The word “parameter” is finally used throughout the manuscript.
- About results: image quality of Figures 1 to 4 and Tables 1 to 2 should be improved. Titles of Table 1 and Table 2 can be shortened by moving the notes in the titles to the end of the tables.
Authors’ response: figures were improved, and titles of table were shortened.
- Discussion: the last paragraph of discussion part can be expanded a little to provide not only the general conclusion but also some detailed information of summary of results.
Authors’ response: A ”Conclusion” were added summarizing the results.

Reviewer 3 Report
In this study, the authors investigated the behavioral consequences induced by aphid resistance in Pisum. This behavioral approach based on the EPG technique follows another study by the authors on the genetic determinism of aphid resistance in Pisum and currently being published in Theoretical and Applied Genetics(TAG). Even if it is difficult to evaluate the originality of this study since the first study published in TAG is not yet accessible, the approach based on the EPG seems to have brought new clues about the physiological mechanisms underlying aphid resistance. The problematic and the hypotheses of this study are clearly exposed and the experimental approach is well adapted. Finally, this study fits perfectly with the scope of this special issue. I have only relatively minor comments on this manuscript.
First of all, I was confused by the term "biomass" (eg . line 115) because I expected this measurement to correspond to the total mass of the aphids after 8 days of development (= the sum of the weights). I then realized that it was not the sum of the aphids but the individual weight of the aphids. I therefore suggest replacing the term "biomass" with "body mass" throughout the manuscript.
Then, a study showed that secondary endosymbionts can modify the feeding behavior of aphids (Leybourne et al 2020 JEB). Authors should specify in the manuscript whether the clones used in this study host secondary endosymbionts.
Finally, I think that the graphs of this study must be improved. First, in Figures 1 and 2, the Y-axis unit should be in parentheses, and the Y-axis range can be increased to allow more room to write the letters above the boxplots. Next, the resolution of Figures 3 and 4 is not sufficient and must be improved to allow correct reading.
Author Response
3rd Reviewer
- First of all, I was confused by the term "biomass" (eg . line 115) because I expected this measurement to correspond to the total mass of the aphids after 8 days of development (= the sum of the weights). I then realized that it was not the sum of the aphids but the individual weight of the aphids. I therefore suggest replacing the term "biomass" with "body mass" throughout the manuscript.
Authors’ response: the suggestion of the reviewer was applied and we replaced the word “biomass” by “body mass”.
- Then, a study showed that secondary endosymbionts can modify the feeding behavior of aphids (Leybourne et al 2020 JEB). Authors should specify in the manuscript whether the clones used in this study host secondary endosymbionts.
Authors’ response: These aphid clones are deprived of any secondary symbionts reported in A. pisum.
- Finally, I think that the graphs of this study must be improved. First, in Figures 1 and 2, the Y-axis unit should be in parentheses, and the Y-axis range can be increased to allow more room to write the letters above the boxplots. Next, the resolution of Figures 3 and 4 is not sufficient and must be improved to allow correct reading.
Authors’ response: All the figures were improved and we hope that they occur in good resolution.
